# Bi-Spectral Infrared Algorithm for Cloud Coverage over Oceans by the JEM-EUSO Mission Program

**DOI:** 10.3390/s21196506

**Published:** 2021-09-29

**Authors:** David Santalices, Susana Briz, Antonio J. de Castro, Fernando López

**Affiliations:** Laboraty of Infrared, Universidad Carlos III de Madrid, Av. de la Universidad, 30, 28911 Madrid, Spain; davidsantalicesmartin@gmail.com (D.S.); decastro@fis.uc3m.es (A.J.d.C.); flm@fis.uc3m.es (F.L.)

**Keywords:** remote sensing, infrared camera, cloud coverage, split-window algorithm, JEM-EUSO

## Abstract

The need to monitor specific areas for different applications requires high spatial and temporal resolution. This need has led to the proliferation of ad hoc systems on board nanosatellites, drones, etc. These systems require low cost, low power consumption, and low weight. The work we present follows this trend. Specifically, this article evaluates a method to determine the cloud map from the images provided by a simple bi-spectral infrared camera within the framework of JEM-EUSO (The Joint Experiment Missions-Extrem Universe Space Observatory). This program involves different experiments whose aim is determining properties of Ultra-High Energy Cosmic Ray (UHECR) via the detection of atmospheric fluorescence light. Since some of those projects use UV instruments on board space platforms, they require knowledge of the cloudiness state in the FoV of the instrument. For that reason, some systems will include an infrared (IR) camera. This study presents a test to generate a binary cloudiness mask (CM) over the ocean, employing bi-spectral IR data. The database is created from Moderate-Resolution Imaging Spectroradiometer (MODIS) data (bands 31 and 32). The CM is based on a split-window algorithm. It uses an estimation of the brightness temperature calculated from a statistical study of an IR images database along with an ancillary sea surface temperature. This statistical procedure to obtain the estimate of the brightness temperature is one of the novel contributions of this work. The difference between the measured and estimation of the brightness temperature determines whether a pixel is cover or clear. That classification requires defining several thresholds which depend on the scenarios. The procedure for determining those thresholds is also novel. Then, the results of the algorithm are compared with the MODIS CM. The agreement is above 90%. The performance of the proposed CM is similar to that of other studies. The validation also shows that cloud edges concentrate the vast majority of discrepancies with the MODIS CM. The relatively high accuracy of the algorithm is a relevant result for the JEM-EUSO program. Further work will combine the proposed algorithm with complementary studies in the framework of JEM-EUSO to reinforce the CM above the cloud edges.

## 1. Introduction

The detection of clouds with airborne instruments is critical when studying the meteorology and climate of the Earth. However, there are other applications where auxiliary systems for cloud detection and characterization are needed, which is the case that concerns us.

Thermography is one of the main techniques for measuring the temperature of things remotely. It is based on InfraRed (IR) cameras to characterize the relationship between the object temperature and the IR energy it emits. Although the first applications of IR cameras were military (World War II), their use spread to many fields since the 1960s. However, the direct application of IR cameras to some applications did not obtain the expected results. Frequently, the lack of knowledge of the physical phenomena involved in the emission, propagation, and measurement of IR radiation led to inappropriate use of commercial IR cameras or prevented a correct interpretation of the images.

To solve a new technological or environmental problem, it is necessary to follow an adequate procedure. The method should begin with a radiometric and spectral characterization of the problem. Afterward, the IR instruments can be developed or adapted according to the previous characterization. Finally, to solve the specific problem, it is necessary to develop specific algorithms that retrieve the physical information from the data provided by the IR instrument: combustion measurements [1,2] forest fires, [3,4] non-destructive analysis, medical applications [5], etc.

Nowadays, technology has evolved significantly, and there are extremely high- performance hyperspectral IR systems. Many of these systems are embedded in satellites and can provide very precise information on many phenomena globally.

However, the most current trend in the space sector is moving towards developing small satellites, nanosatellites, or CubeSats. At present, many environmental and technological problems do not require the high spectral resolution that current technology offers. However, they do require simple systems with low cost, low mass, and low energy consumption. In this framework, it makes sense to recover bi-spectral thermography systems, which are simpler and cheaper than sophisticated multispectral systems. The spectral and radiometric characterization of the problem should be the basis to simplify the devices. This alternative approach facilitates observation at higher spatial and temporal resolution and addresses monitoring problems in specific areas. This is also the case of some instruments using ancillary devices to provide complementary but essential information.

Precisely, the work that we present here aims to develop and validate an algorithm to detect clouds from the information provided by a simple bi-spectral IR camera. In the Joint Experiments Missions- Extreme Universe Space Observatory (JEM-EUSO), a secondary instrument (a bi-spectral camera) will carry out the detection of the clouds. However, the relevance of the results we present here goes beyond the JEM-EUSO missions. New instruments for mini and nanosatellite constellations, mainly focused on communications and Earth observation, could use this algorithm.

The objective of the international JEM-EUSO program is to observe Ultra High-Energy Cosmic Rays (UHECRs) to explain the origin and nature of such particles [6]. When a UHECR collides with an atmospheric nucleus, it causes an Extensive Air Shower (EAS), which is a cascade of charged particles throughout the atmosphere. The charged particles excite nitrogen molecules and we can detect the ultraviolet (UV) radiation produced in this process (fluorescence and Cherenkov radiation). The analysis of the UV images will give information on the UHECR properties. That is the observational principle of all the experiments of the JEM-EUSO program. However, the flux of these particles is extremely low (a few per km${}^{2}$ per century at extreme energies such as *E* > 5 × 10${}^{19}$ eV). The JEM-EUSO Collaboration has addressed that challenge with space observatories. The final objective of the JEM-EUSO program is to realize a space mission with a super-wide-field telescope. It will look down from space onto the night sky to detect UV photons emitted from the EAS generated by UHECRs in the atmosphere. This is also the measuring principle of other pathfinders boarded on balloons and satellites to test the technologies involved in this ambitious program [7,8].

However, the presence of clouds between the EAS and the UV telescope may interfere with the EAS measurement. It may also lead to a misinterpretation of the observations. Then, for this type of application, an auxiliary instrument that maps the state of the clouds when a UHECR occurs is essential.

Today there are many weather satellites with instruments capable of determining the state of the sky with great accuracy [9,10,11]. Most of them base their operation on a massive study of bands that allow an exhaustive atmosphere characterization, e.g., MODIS includes 36 spectral bands ranging from 0.4 to 14.4 μm. The Advanced Very High-Resolution Radiometer (AVHRR) is a multispectral sensor with six spectral bands included in NOAA satellites since the last 1970s [12]. Even the Imaging Infrared Radiometer (IIR), which is part of the payload of the Cloud-Aerosol Lidar and IR Pathfinder Satellite Observation (CALIPSO-NASA), provides information in three spectral bands [13]. The Spinning Enhance Visible and Infrared Imager (SEVIRI) in MSG (Meteosat Second Generation) satellites observes the Earth in 12 spectral bands [14]. Moreover, the Meteosat Third Generation Sounder (MSG-S) includes an Infrared Sounder (IRS) based on an imaging Fourier interferometer with a hyperspectral resolution of 0.625 cm${}^{-1}$ [15]. Logically the performance of these sophisticated instruments has superseded the simple bi-spectral systems since the 1990s [16,17,18].

However, current weather satellites do not have the temporal and spatial resolution necessary to ensure simultaneous information to JEM-EUSO main instrument. In addition, the high cost, weight, power consumption, and dimensions of those high-performance spectroradiometers make them non-viable for auxiliary purposes of JEM-EUSO missions.

For this reason, the JEM-EUSO instrument and some of the pathfinders include in their payload a bi-spectral camera in the thermal infrared (TIR) spectral region [19,20,21,22].

Reference [19] includes detailed information on the specifications of the IR camera of the initial JEM-EUSO mission as an example. Table 1 summarizes the main characteristics of weight, dimensions, consumption, and spectral bands of the IR camera. For comparison purposes, the table includes the same information for the Moderate-Resolution Imaging Spectroradiometer (MODIS), a multispectral instrument of reference in the field of the Earth observation and onboard the Terra and Aqua satellites. Other JEM-EUSO missions also propose IR cameras of similar characteristics.

The JEM-EUSO camera has its bands located around 11 and 12 μm for different reasons: the requirement of measuring at night, the 8–13 μm atmospheric windows, and the slight differences in cloud absorption between these bands [23]. It is also important to note that using two adjacent bands simplifies the instrument since it only requires one detector array.

**Table 1 sensors-21-06506-t001:** Specifications of a initial IR camera on board JEM-EUSO instrument [19] and MODIS instrument [24].

	JEM-EUSO Camera	MODIS
Mass (kg)	11	228.7
Dimensions (m)	0.40 × 0.40 × 0.37	1.0 × 1.0 × 1.6
Power consumption (W)	15	162.5
Data Rate (Mbps)	0.04	10.6
No. of bands	2	36
Band #11 (μm)	10.3–11.3	10.78–11.28
Band #12 (μm)	11.5–12.5	11.77–12.27

Concerning the data analysis, most of the cloud detection methods use radiometric and multispectral single-pixel tests, which rely on selecting different thresholds [10,25,26] or textural and nearby pixel measures [27,28].

The use of radiometric methods focused on cloud detection is widespread and exploits the information of those multiple bands. The Cloudiness Mask (CM) used by MODIS uses 11 spectral tests for 19 different bands [25]. The cloud detection algorithm of the second generation Meteosat involves 12 different bands [10]. The AVHRR employs five different channels involved in five spectral tests and two spatial tests. Although those satellites use a wide variety of bands to determine the CM, they include a few tests that only use the 11 and 12 μm bands (e.g., a gross 11 band test or a thin cirrus test that uses both bands).

At this moment, the most innovative methods are learning-based techniques such as a machine or deep learning [29,30,31,32,33,34,35]. Nonetheless, learning-based methods are not easy to apply to JEM-EUSO missions.

Since the JEM-EUSO IR cameras have slight differences in their designs (resolution, noise, altitude, latitude, etc.), it would be necessary to readapt and retrain the learning-based methods for the different pathfinders. In addition, the IR systems onboard pathfinders do not provide enough data to train those methods due to the short duration and trajectory of the flights. For this reason, we have developed a radiometric algorithm that can be more suitable in the frame of the JEM-EUSO missions.

However, the bi-spectral concept of JEM-EUSO IR cameras adds strong restrictions to the available tests when determining the CM for this system. Nevertheless, it is still possible to retrieve information about cloud coverage using those bands. Moreover, the conditions where an EAS can be measured make the development of the CM easier. JEM-EUSO program will detect the EAS just when the UV background noise is low enough. Clouds, then, only have to be detected during night-time and mainly over oceans. We have extended the study to day-time images to interpret the results better and make this new CM more applicable.

The goal of the study we present is to design and evaluate a CM test to determine the presence of clouds in the Field of View (FoV) of the JEM-EUSO telescope, over oceans and during night-time. Although we have also studied the performance of a gross test based only on the brightness temperature measured at 11 μm, our proposal is finally a bi-spectral CM due to its better results. Our CM uses brightness temperatures (BTs) measured in the bands centred at 11 and 12 μm (from now on $BT_{11}$ and $BT_{12}$) along with the Sea Surface Temperature (SST) as ancillary data.

The main idea of the methodology we propose is to establish a relationship between the $BT_{11}$, the Brightness Temperature Difference (BTD) between $BT_{11}$ and $BT_{12}$, and the ancillary SST using a set of real images in $BT_{11}$ and $BT_{12}$ bands. The objective is to get a statistical estimation of the $BT_{11}$ as a function of the BTD and SST for clear-sky pixels. Since the BT of cloudy pixels is lower than that of the clear pixels, the difference between the real and estimated $BT_{11}$ can be useful to determine the pixel state. Based on this difference, we define a threshold using a similar technique as in [36] to determine whether the pixel is clear or cloudy.

The use of two adjacent bands has been used before in other CM algorithms [10,25,36]. The methodology we propose combines the advantages of those CMs but also provides several novel ideas. First, in this work, the estimate of the $BT_{11}$ for clear sky pixels is calculated from a statistical analysis of thousands of images to consider all the atmospheric scenarios. Therefore, the $BT_{11}$ values do not depend on the performance of any radiative model or the estimate of the atmospheric conditions. Second, since our original data are $BT_{11}$ values, the CM threshold is calculated on the $BT_{11}$ value not on the SST one, as in [25] and [36]. Third, the threshold is not determined directly from the intersection between the distribution functions of the clear and cloudy pixels but by optimizing the results using skill scores, as explained in Section 2.3. Finally, we include a specific analysis of partially cloudy pixels.

Although there are some pathfinders, including IR cameras, for example, the EUSO-SPB II (EUSO-Super Pressure Balloon II), the IR cameras have not provided enough images [22]. For that reason, MODIS data have been used in this work, allowing us to develop our algorithm and validate it. Considering this CM is based on a split-window algorithm test, we have called it the Split-Window Cloudiness Mask (SWCM).

As JEM-EUSO does not retrieve the SST, a global SST model is needed. In this study, the NOAA 1/4${}^{\circ }$ daily Optimum Interpolation Sea Surface Temperature (or daily OISST) [37] provides the SST estimation.

Section 2.1 contains a short description of the data used to design and validate the algorithm. Section 2.2 describes the fundamentals of the algorithm. Section 2.3 details the methods to calculate the thresholds and coefficients that define the SWCM. Afterward, the results of our methodology, that is, the final coefficients and thresholds that define the SWCM and its validation, are presented in Section 3. In Section 4 we discuss the results attained in the previous section and compare them with those of other authors. The last section, Section 5, provides a summary of the methodology and the main results of this work. It also includes the relevance of the results in the framework of JEM-EUSO and remote sensing in general. In this section, we also point out some future working lines oriented to combine tests of different nature in a more global CM to improve the results of individual single-test CMs.

## 2. Materials and Methods

In this section we will describe the data used to design and validate the SWCM. Afterwards, the theoretical basis of the SWCM is summarized and the SWCM is defined. The last subsection details the methodology to optimize the thresholds that characterize the SWCM.

### 2.1. Data

Since the CM will be applied in the future to JEM-EUSO systems, it must be based on the data that provide those systems. Although the spectral characteristics of the different JEM-EUSO IR cameras can be slightly different, the bands of the IR reference instrument are centred at 10.8 and 12.0 μm. As 31 and 32 MODIS bands are centred at 11.030 and 12.020 μm, we have selected the brightness temperature data at these two bands to design our CM algorithm. In addition, MODIS products include accurate cloud features, which are crucial to analyze and evaluate the proposed algorithm.

In this work, we use daily Collection 6.1 level 2 MODIS images [38] combined with a global SST model. Table 2 summarizes the data fields utilized in this study.

#### 2.1.1. MODIS Data

Collection 6.1 level 2 from MODIS (MYD06) is a product that contains processed information about the optical properties of the clouds and other processed MODIS products (as the MODIS CM).

MODIS products have been widely used as ground truth to evaluate new CM algorithms due to his great confidence [11,39]. A comparison between the Active Remote Sensing of Clouds (ARSCL) and the MODIS CM presents an agreement of 85% between both techniques [9,40]. The largest difference between the two methods occurs for high and thin clouds.

The similarity between JEM-EUSO IR camera bands, its global coverage, and its high precision; make MODIS a great choice to calculate and validate the SWCM.

The data fields taken from MODIS include $BT_{11}$, $BT_{12}$, the MODIS Cloud Fraction (CF), and geolocation images. Although MODIS bands have, at least, resolutions better than 1 × 1 km${}^{2}$, $BT_{11}$, $BT_{12}$ are provided at 5 × 5 km${}^{2}$. For that reason, in this work, the 5 × 5 km${}^{2}$ is the spatial resolution used. Additionally, most MODIS cloud products have that resolution.

However, although MODIS provides a CM with a resolution of 5 × 5 km${}^{2}$, we have used the MODIS CF product also at a 5 × 5 km${}^{2}$ resolution. The main reason is that MODIS 5 × 5 km${}^{2}$ CM is subsampled from the MODIS 1 × 1 km${}^{2}$ CM [41]. The value of the 5 × 5 km${}^{2}$ CM is only representative of the central pixel of the corresponding area at 1 km resolution and not of the general cloudiness state of the 5 × 5 km${}^{2}$ area.

The CF product is immediately derived from the MODIS 1 × 1 km${}^{2}$ CM and is defined as the percentage of the cloudy pixels in a 5 × 5 km${}^{2}$ pixel and then reflecting better the state of the pixel. Therefore, the CF product is more representative than the CM at 5 × 5 km${}^{2}$ resolution.

#### 2.1.2. Ancillary SST

As SST input, we use the daily OISST. Numerous algorithms use this kind of OISST products as input, e.g., MODIS uses a weekly OISST to calculate initial radiances. The OISST is correlated with the MODIS data fields using the geolocation that both products provide. Since the resolution of the SST model is worse than that of MODIS images, we have applied a linear interpolation to correlate MODIS and OISST products.

#### 2.1.3. Region of Study

Since the final JEM-EUSO instrument is thought to be on board the International Space Station (ISS), the trajectory does not include polar latitudes larger than 60${}^{\circ }$. Thus, a total of 6000 midlatitude and 5500 tropical images belonging to 2018 have been selected to represent all possible scenarios of the JEM-EUSO IR camera, considering the latitudes and seasonal variations involved in the trajectory of the ISS. To avoid using cross data between the method and validation, we have split MODIS data into two different groups: the methodology group (5704 images) and the validation group (3557 images).

### 2.2. SWCM Definition

The main idea of the methodology we propose is to establish a relationship between the $BT_{11}$, the BTD and the ancillary SST using a set of real images. The objective is to get a estimation of the $BT_{11}$ as a function of the BTD and SST for clear-sky pixels. The difference between the real and estimated $BT_{11}$ can be useful to determine the pixel state. Based on this difference, we define a threshold to determine whether the pixel is clear or cloudy.

Figure 1 shows a flowchart of the SWCM.

#### 2.2.1. Theoretical Basis

Due to the close relationship between the SST and the BT or radiance observations in the TIR spectral region, some authors have introduced several algorithms to find the SST from radiative sources [42]. On the contrary, using a guessed surface temperature to calculate cloud properties is also not uncommon. The CM presented in [11] uses simulated real-time clear-sky IR radiances for different tests. Numerical Weather Prediction data supply the atmospheric parameters required for those simulations. The SST values are also needed.

The combination of [43,44,45] lead to the Equation (Equation 1) that allow us to estimate the $BT_{11}$ value from the BTD and the SST:(1)$$\begin{array}{cc} \; & BT_{11,e}=A\cdot SST\\ \; & +BTD\left(B_{1}+B_{2}\cdot SST\right)+C\cdot \left(1-\sec\left(\theta \right)\right)BTD+D \end{array}$$
where $BT_{11,e}$ is an estimation of the clear-sky $BT_{11}$; θ is the zenith angle and *A*, $B_{1}$, $B_{2}$, *C* and *D* are constants to be determined.

In this work, we also propose the use of an ancillary SST global model together with the radiative information from a bi-spectral system to estimate the $BT_{11}$ value. However, we determine the coefficients of Equation (Equation 1) through a regression fit and not calculated from radiative simulations as in [10]. Since the coefficients depend on the conditions of the scenario (midlatitude/tropical), we calculate two different sets of coefficients for each one. Figure 1 shows the stage at which the flowchart makes the selection of these coefficients using the geolocation (box Select Latitude Model).

In the next section, we explain how the SWCM uses the difference between the real and estimated $BT_{11}$.

#### 2.2.2. SWCM Procedure

To obtain the SWCM, we define a new parameter, $\Delta BT_{11}$, which measures the difference between the real $BT_{11}$ and the estimated $BT_{11,e}$ (for clear sky).
(2)$$\Delta BT_{11}\;=\;BT_{11}-BT_{11,e}$$

If the radiative properties of the atmosphere remained the same, clear-sky pixels would satisfy $BT_{11,e}\;=\;BT_{11}$. However, due to the atmosphere’s variation, the $BT_{11}$ is not the estimated one. Thus, we need a threshold to establish the $BT_{11}$ range of clear-sky pixels.

Equation (Equation 2) determines the difference between the real and a estimated $BT_{11}$ (values of $\Delta BT_{11}$ close to zero show a high probability of being clear sky). The definition of a probability threshold for $\Delta BT_{11}$ will allow generating a binary cloudiness mask. The SWCM is defined as:(3)$$SWCM\;=\;\left\{\begin{array}{cc} Cloudy & \mathrm{if}\;\Delta BT_{11}<\tau \\ Clear & \mathrm{if}\;\Delta BT_{11}\geq \tau \end{array}\right.$$
being τ a threshold to determine. In this study, we calculate different τ for different conditions (tropical/midlatitude or day/night). Figure 1 shows the stage where the SWCM makes this selection (box Select Threshold).

### 2.3. Methods

The method used to determine the coefficients consists of:Selecting the MODIS images from the method group.Separating the images for two different regions: tropical and midlatitude.Fitting each subset of clear-sky pixels to the Equation (Equation 1) to obtain the coefficients.

The method used to determine the thresholds consist of:Determining $BT_{11,e}$ for each pixel using the coefficients of the Equation (Equation 1).Determining $\Delta BT_{11}$ for each pixel.Determining a ground truth CM to compare with the SWCM.Grouping the pixels of the images for the different scenarios (tropical/midlatitude, day/night).Scanning different τ thresholds for $\Delta BT_{11}$ to discern the cloudiness state (clear/cloudy sky).Selecting the optimum τ threshold for each scenario.

### 2.4. Coefficients Determination

Although other works base the calculation of the coefficients on direct radiative models, in this work, we determine the coefficients of Equation (Equation 1) empirically through a robust regression fit, using a bi-square weighting. This regression involves only pixels with a CF equal to zero, securing that no clouds were involved in the fitting. All pixels containing land or sea ice were also discarded.

A total of 4006 midlatitude (|latitude| > 23.44 & |latitude| < 66.56) and 3661 tropical (|latitude| ≤ 23.44) MYD06 images were involved in the fit. The images were selected in an attempt to cover evenly all oceans and all seasons of the year, as well as the different hours of the day.

### 2.5. Threshold Determination

Since the SWCM is a categorical variable obtained from continuous data, it depends strongly on whether or not that continuous variable exceeds a specified threshold, τ.

Then, to obtain the SWCM, an optimal threshold for the $\Delta BT_{11}$ parameter has to be defined. In this study, we perform a comparison between MODIS CF derived CMs and the SWCM for different τ values to obtain the optimal threshold. To quantify those comparisons, we use diverse skill scores, that is, statistical measures of the accuracy and performance of a classifier. A common technique to evaluate categorical classifiers such a CM [11,39] are the use of skill scores [46]. This study involves five different skill scores: the Proportion Correct (PC), Frequency Bias (FB), Probability of Detection (POD), and the Kuiper’s Skill Score (KSS) (Appendix A contains the definitions and use of these skill scores).

Figure 2 represents the probability density functions of $\Delta BT_{11}$ for totally clear (CF = 0), totally cloudy (CF = 100), and mixed pixels (0 < CF < 100) for midlatitude images. Totally clear pixels (continuous line) are grouped around $\Delta BT_{11}\;=\;0$, within the range of −3 K and 3 K. That range also embraces some totally cloudy pixels (dotted line). The small overlap existing for totally clear and cloudy pixels means no threshold splits both classes entirely. The mixed pixels (dashed line in Figure 2) increase the overlapping for values between −10 and 3 K. That range also embraces some totally cloudy pixels (dotted line). The easiest solution would be to reject those mixed pixels by applying spatial and textural techniques, such as the Sobel algorithm [47]. However, the number of pixels discarded would be too high, at least at 5 × 5 km${}^{2}$ spatial resolution. For this reason, this study includes mixed pixels. The optimal τ value is the one for which the SWCM gives the best results.

Since the ground truth used in this study is the MODIS CF and not MODIS CM, the evaluation of the binary SWCM against the 0–100% MODIS CF is not direct. On the contrary, it means that a new parameter is necessary, *h*. CF above the *h* parameter leads to cloudy pixels, and those below will be considered clear sky. This new binary mask calculated from the MODIS CF product can be used as ground truth to calculate the optimal τ value and validate the SWCM performance.

Therefore, two kinds of thresholds are necessary for this study. The *h* parameter used to generate the ground truth binary mask from the MODIS CF product and τ optimal threshold used to define the SWCM based on the $\Delta BT_{11}$ parameter.

#### 2.5.1. Ground Truth Cloud Masks

In order to evaluate the impact of these mixed pixels, we have considered two ground truth CM derived from MODIS CF. In this sense, we have evaluated the SWCM from a twofold point of view.

The first ground truth CM is the Reference Cloud Mask (RCM), which gives an idea of the SWCM performance in real situations where mixed pixels are always present. A transformation of the CF into a CM using the *h* parameter conducts to the RCM:(4)$$RCM\;=\;\left\{\begin{array}{cc} Cloudy & \mathrm{if}\;CF>h\\ Clear & \mathrm{if}\;CF\leq h \end{array}\right.$$

In principle, the h parameter can be chosen at the user’s convenience and depends on the final application. To avoid an arbitrary selection of *h*, in this paper, *h* has also been selected based on skill scores measures.

In this framework, the best response to the cloud mask determines the threshold. The Receiver Operational Characteristics (ROC) curves are one of the most common methods to evaluate that response [48]. ROC curves show the relationship between the PODs of classes of a classifier, in this case, between POD_clr_ and POD_cld_. By computing ROC curves for different *h* values, we can choose the optimal *h* threshold. Figure 3 represents the ROC curves of different *h* values. The optimal *h* parameter is 40% since it is the one that maximizes the area under the curve.

In addition, we introduce the Pure Cloud Mask (PCM) to evaluate the SWCM capacity to discriminate totally clear and totally cover-sky pixels. To achieve that, the PCM excludes all pixels with a CF distinct to 0 or 100%, avoiding the mixed pixels. In this way, it is possible to determine if the discrepancies are due to the methodology or the mixed pixels.

Figure 3 also depicts the ROC curve corresponding to only completely cloudy and clear pixels, named Pure Cloudiness Mask (PCM). This curve shows that the proposed cloud mask performs as a near-perfect classifier when no mixed pixels are considered.

#### 2.5.2. Optimal τ Threshold Calculus

We can determine the optimal τ threshold after generating the ground truth CMs (RCM and PCM). For this purpose, we calculate the SWCM for different τ values. Then, we compute the skills scores mentioned in Appendix A by comparing the SWCMs with the ground truth CMs.

Figure 4 represents the PC, KSS, and FB in function of the τ threshold value.

The PC is not the best skill score for unbalanced problems (where a class is over-represented). In addition, high PC values appear even though the predictor shows no skill. e.g., for τ > 4 K, all clear sky pixels get classified as cloudy ones (Figure 2) but the PC is still high for those values (Figure 4).

Compared with the PC, the KSS measures the ability to separate both categories and solves that inconvenience. In Figure 4, we can observe that for the maximum value of the KSS, the FB is close to one, which means that no one class is over-represented.

Since the KSS seems to be the best choice for this problem, we have calculated the optimal τ threshold by maximizing the KSS (Figure 4).

## 3. Results

The results of this work have been split into two subsections: Section 3.1 specifies the results of the proposed methodology, that is, the coefficients and thresholds of the SWCM. Section 3.2 includes the results of the validation against MODIS multiband algorithm.

### 3.1. Cloud Mask Results

#### 3.1.1. Cloud Mask Coefficients

The CM coefficients have been calculated for the tropical and midlatitude models and are summarized in Table 3. Both models present similar coefficients except for $A_{1}$ and $B_{2}$, associated with the BTD.

#### 3.1.2. τ Threshold

We have calculated the optimal value $\tau_{opt}$ for mid and tropical latitudes, during the day and night time and for the PCM and RCM, obtaining a total of eight different thresholds, summarized in Table 4.

### 3.2. Validation

The validation procedure consists of applying the SWCM to a set of 3800 MODIS images. The results are compared with those of the PCM and RCM described in the previous section (Section 2.5.1). The two kinds of validation allow us to analyze the cloud fraction effect on the SWCM (Section 3.2.1).

The validation set contains images of each month of the 2018 year during the day and night-time, including latitudes between 0 and $60^{\circ }$ degrees, which makes possible an analysis of the day/night influence and the seasonal variations (Section 3.2.2 and Section 3.2.3 respectively).

#### 3.2.1. Validation Results

Around 95 × 10${}^{6}$ tropical and 155 × 10${}^{6}$ midlatitude observations were used to perform the validation. This number is inferior when the PCM is used as ground truth since mixed pixels got excluded.

Table 5 presents the contingency values (*a*, *b*, *c*, and *d*), the PC, KSS, and FB of the SWCM. The table summarizes the tropical and midlatitude case. The skills scores have been calculated using the two different ground truths, PCM and RCM. Table 5 includes the data during the day and night-time conditions separately and merged.

Table 6 illustrates the monthly variation of the skill scores. When the ground truth is the PCM, the PC and KSS is 0.97–0.98 and 0.90–0.97, respectively. The POD for the clear-sky class is 0.93–1.00 and 0.97–0.98 for the cloudy one. The similarity between both PODs values reflects the unbiased nature of the selected thresholds. The close to 1 value of the FB indicates that almost no bias is introduced compared to MODIS for the totally-clear and cloudy states.

When the ground truth is the RCM, the PC and KSS decrease to 0.90–0.92 and 0.83–0.84, respectively. The POD is lower for both classes, being 0.88–0.92 for cloudy pixels and 0.91–0.96 for clear-sky pixels. The FB is 0.90–0.93 for the cloudy class and 1.22–1.33 for the clear-sky class, meaning that the SWCM is biased in comparison to the RCM, overestimating the clear sky pixels and underestimating the cloudy ones.

As an example Figure 5 represents a comparison between the RCM and the SWCM on 1 January 2018 at 01:15 UTC. The image on the left represents the RCM and the image on the right shows the discrepancies between the RCM and the SWCM. In general, the agreement between both CMs is quite good, and the most significant differences are on the edges of the clouds (discussed in Section 4).

#### 3.2.2. Day or Night Influence

Since the data include pixels during the day and night-time, Table 5 includes those conditions separately.

The results show no significant differences between the day and night time skill scores, excluding results in midlatitude when RCM is used as ground truth, with better results during day-time with a KSS of 0.86 (0.78 for the night-time).

#### 3.2.3. Seasonal Variation

As commented in Section 2.1.3, the MODIS images have been selected to be representative of the different seasons. Table 6 summarizes the obtained KSS, PC, POD, and FB for each month.

Although some skill scores remain almost stable (the PC and the POD for cloudy pixels), some scores show a monthly variation (the POD for clear-sky pixels).

Figure 6 shows the monthly variation of the POD for clear-sky pixels for midlatitude north, midlatitude south, and tropical regions. The tropical POD_clr_ remains stable, but for midatitude, the POD_clr_ is worst for the winter-spring seasons with values of 0.76 (in comparison to summer-autumn months, with values of 0.98).

The lower POD for clear-sky pixels during winter-spring months causes a monthly variation of the KSS and FBs.

## 4. Discussion

As seen in the Section 2.3 and Section 3.2, different thresholds τ and different performances are derived depending on the day/night state, the latitude, or season. In this section, we will discuss these results and compare them with those of other authors to better evaluate our SWCM.

As expected, the comparison with the PCM ground truth, which evaluates the performance of the SWCM to classify clear and cloudy pixels, gives more satisfactory skill scores than the comparison with the RCM ground truth, which better reflect the overall skill of the SWCM including mixed pixels. For the PCM, the PC, KSS, POD, and FB take values close to one (the perfect score for those skill scores). That means that the SWCM is able to classify clear and cloudy pixels with high accuracy and that the mixed pixels are the primary source of error in the SWCM.

In general, we can say that the results are very encouraging since the PC is always higher than 0.90 (including mixed pixels) for all the latitudes regardless of whether it is day or night (Table 5). The difference in the total PC score when using both ground truths is 0.08 (Tropical) and 0.07 (midlatitude), which means that the performance of the SWCM is also good in realistic scenarios, including mixed pixels.

The KSS presents the same behavior, with a total difference of 0.13 (tropical) and 0.7 (midlatitude). This fact reveals that the optimization of the h threshold (h = 40%) solves the issue of the $\Delta BT_{11}$ overlapping between the mixed pixels and the clear and cloudy ones, at least partially. As an illustrative example, Figure 7 represents the probability density functions of the scene represented in Figure 1 divided in pixels with CF below 40% and pixels with CF above 40%. In general, those mixed pixels are located in cloud edges. Nevertheless, the influence of those pixels in the SWCM performance is strong because of the size of the pixels (5 × 5 km${}^{2}$) that increases the percentage of pixels partially covered. Therefore, it is expected that the SWCM performance improves when applied to better spatial resolutions [49] (as JEM-EUSO systems).

As mentioned before, the $BT_{11,e}$ and the τ depends on the blackbody emission of the ocean (according to its temperature) and the absorption due to the atmospheric water vapour content. However, $BT_{11,e}$ and $\tau_{opt}$ calculus uses not a real SST but a daily SST provided by the NOAA Daily OISST global model. The OISST is constructed by combining observations from different platforms (satellites, ships, buoys, and Argo floats). For this reason, the OISST temperature values can differ from those of a remote sensor (±0.5 K on average [50]). Nevertheless, if there is a bias between both temperatures for any condition (i.e., different CF conditions as shown in [51]) the effect of those differences is minimized in the statistical procedure, that is, in the fitting process to calculate the coefficients of Equation (Equation 1) and in the thresholds optimization.

In addition, the small diurnal SST oscillation [52] could entail some errors that have been avoided calculating two different τ for day and night (Table 4). The difference between both τ is higher for Tropical latitudes, where the SST daily oscillation is also higher. The negligible difference between the day and night values of all the scores in Tropical analysis (Table 5) indicates that the diurnal variation is properly taken into account using different τ for day and night. The midlatitude images analysis shows the same behavior except for the values of POD_clr_ for which there is a clear difference between day and night. This exception could be related to the well-known difficulty of identifying clear pixels when the underlying surface is at low temperature, and there is no good contrast with clouds, which are also cold. This poor contrast between the cold SST and the clouds could also explain the seasonal variation in the midlatitude region observed in the validation section (Section 3.2, Table 6 and Figure 6).

The differences between day and night performance could be associated with the variation of the atmospheric state as well. Precisely, the atmospheric vertical profiles also undergo a seasonal variation, especially the water vapor profile, which has the main responsibility for the atmospheric absorption in the TIR. Since we do not perform regression fits for each month in each geographical region, the methodology averages the atmospheric seasonal variations. As the water vapor variation occurs in the firsts kilometers of the atmosphere, it affects the POD_cld_ to a lesser extent because the clouds above the lower layers of the atmosphere can shield what happens below them. This explanation is in agreement with a lower difference between day and night POD_cld_ compared to the POD_clr_ one in the midlatitude region (Table 5). It also agrees with a lower seasonal variation in POD_cld_ compared to POD_clr_ for the same region, as can be seen in Table 6. However, the tropical region does not show this seasonal variation, as the water vapor content almost does not change throughout the year. Finally, the different proportions of clear and cloudy sky pixels existing could also cause the seasonal variation of the PODs in the midlatitude region, e.g., for the validation subset of images, for midlatitude north, during august, the ratio between totally cloudy and clear pixels is 4.1; meanwhile, during march month, the ratio is more significant, with 6.3.

To summarize, we find slightly better results for Tropical than midlatitude regions (Table 6). However, the skills scores are very similar between the tropical and the midlatitude cases during summer-autumn months (the discrepancies occur during the winter-spring months). In almost all cases, the PC is more significant than 0.90, and the KSS is higher than 0.82 (Tropical) and 0.69 in the worst case of midlatitude.

In any case, to determine the scope of the SWCM in a more general framework, our results have been compared with those of other authors who also use MODIS CM as ground truth. Table 7 contains the skills scores of different CMs when compared to MODIS CM. It should be mentioned that these comparisons do not take into account that these works cross the data of two different satellites and instruments, increasing the difference with the MODIS CM. On the contrary, this work compares the same MODIS instrument’s data to focus only on the algorithm and CM performance.

The SWCM test has been compared with a $BT_{11}$ gross test ($BT_{11}$ > threshold). The results (Table 7) show that the results of the test used are better than the $BT_{11}$ gross test. This difference becomes greater for midlatitude, where the classification of the $BT_{11}$ gross test is poor.

The INSAT-3D Gaussian Mixture Model (GMM) CM [39] is an algorithm that uses two TIR channels and one Middle IR (MIR) channel. It is based on the assumption that cloud data radiances are clustered in different Gaussian distributions. The CM is obtained then by merging those clusters into cloudy and clear classes.

In [11], an operational CM for the Advanced Geostationary Radiation Imager (AGRI) on board Fengyun-4A is presented. The algorithm applies 13 spectral and spatial tests based on six different bands, producing a four-level CM product. The algorithm makes use of simulated real-time clear-sky IR radiances, using data from the Global Forecast System.

Even though the SWCM uses only one test based on two TIR bands, its global PC (0.91) is similar to the PC values of the other CMs. It is higher than the INSAT-3D GMM PC (0.76), although lower than the Advanced Himawari Imager (AHI) PC (0.93).

Concerning the POD, the SWCM POD${}_{Cld}$ is of the order of the others, but the SWCM POD${}_{Clr}$ is the highest. The SWCM POD${}_{Clr}$ is higher than the SWCM POD${}_{Cld}$, unlike in the other CMs. It is noticeable that the SWCM KSS score, which measures the ability to separate clear and cloudy pixels, is the highest one.

When analyzing the FAR, the SWCM FAR${}_{Cld}$ is lower than the others, although the FAR${}_{Clr}$ is higher. Nevertheless, in general, all the FARs are in the same range.

Other works that show similar POD scores can be found in [53].

To summarize, the performance of the SWCM is similar to the one of the other CMs based on multispectral or spatial tests, even though the SWCM is a one-test CM based on only two spectral bands.

## 5. Summary and Conclusions

The main objective of this new SWCM is to provide the JEM-EUSO program with a cloudiness map in the interest regions. For this reason, we have only focused on areas over oceans because the UV principal instrument will keep clear of populated areas to avoid the associated UV light pollution.

Our proposal is a simple cloud mask test based on a split window algorithm. The inputs of the SWCM are the BTs in two spectral bands in the thermal infrared region (centered at 11 and 12 μm) and SST data provided by a Global Model OISST. The method is based on two stages. The first one is an estimate of the brightness temperature of clear sky at 11 µm. The statistical procedure to calculate the estimate of $BT_{11}$ is one of the novel contributions of this work. The second step calculates the difference between the estimate $BT_{11}$ and the measured one. To classify the pixel, that difference is compared with a threshold. The procedure to calculate the thresholds is also novel.

The usefulness of this SWCM in the framework of JEM-EUSO is twofold. On the one hand, it will determine if there are clouds in the FoV of the UV main telescope. On the other hand, it will select the cloudy pixels where the radiative algorithms will be applied to retrieve the cloud top height, which is the main objective of the JEM-EUSO IR Camera [54].

Concerning the resilience of the SWCM, despite the IR cameras of the different JEM-EUSO missions having slight differences in their designs (resolution, noise, altitude, etc.), the SWCM could be applied to all the JEM-EUSO pathfinders. The spectral bands (centering and width) are very similar to each other and very similar to the MODIS spectral bands. For this reason, the results obtained by applying SWCM to MODIS data can be extrapolated to the IR systems of the JEM-EUSO missions and other systems with similar spectral bands. The proximity between the bands and their coincidence with an atmospheric window also favor extrapolation. Finally, the SWCM uses brightness temperature data, which is calibrated and does not depend heavily on the sensor used.

Since the JEM-EUSO instrument is not already in orbit, we have used MODIS data. Precisely, MODIS bands #31 and #32 are very similar to those of the IR JEM-EUSO Camera. MODIS data have allowed us to develop the SWCM and validate it without crossing data between different instruments and, therefore, without introducing instrumental effects in the evaluation of the CM performance. However, in the future, these instrumental differences will have to be studied.

The SWCM results are very good when applied to totally cloudy or clear pixels, which implies that the proposed algorithm is an appropriate classifier. Comparing the results using PCM and RCM reveals that the main discrepancies are related to partially cloudy pixels at the cloud edges. However, they have been minimized by optimizing the h threshold. However, as the spatial resolution of JEM-EUSO (about 0.75 × 0.75 km${}^{2}$) is better than the one used in this work, the expected results in the JEM-EUSO IR camera should be better. Nevertheless, those mixed pixels could also be identified and/or discarded to improve the algorithm accuracy by applying some spatial and textural techniques. The final solution will depend on the JEM-EUSO requirements.

The lower performance corresponds to the POD_clr_ for midlatitude regions during the winter months. These results could be improved by conducting specific regression fits and calculating new coefficients *A*, $B_{1}$, $B_{2}$, *C*, and *D* for that period in midlatitude regions. Moreover, if the JEM-EUSO mission detected an EAS, local coefficients could be calculated through the radiative Transfer Equation [11], using vertical water vapour profiles provided by numerical weather prediction models such as WRF or GFS [55].

The scope of our study has been determined by comparing SWCM with other relevant studies. The comparison between the skill scores (Table 7) allows us to assert that the SWCM presents a performance similar to other studies that use MOD35 data as ground truth, even though the SWCM uses just one test based on only two spectral bands.

Nevertheless, in the future, it is expected to integrate this algorithm in a more elaborated CM algorithm, also using other complementary tests. Actually, other works based on spatial analysis [54], deep learning [56], or Numerical Weather Forecast models [57] have already been carried out within the JEM-EUSO community.

It is also important to emphasize the relevance of our results in the field of future remote sensing sensors. The great advantage of the proposed SWCM algorithm is its easy hardware implementation. It requires defining only two spectral bands by using band-pass interferential filters on two arrays of the same detector material. The use of only one type of material means a big simplification in terms of electronics and data acquisition and management.

Finally, we would highlight that this algorithm could also be applied to data of plenty of satellites, both operative and non-operative since the use of the TIR bands has been widespread since the second half of the last century. Moreover, the concept proposed in this article can be easily transferred to nanosatellites and CubeSats constellations devoted to Earth observation, following the current trend of development of simple and low cost, mass, and energy-consumption systems. The possibility of monitoring the presence of clouds at high resolution and in specific areas with simple systems will allow providing complementary and valuable information for numerous environmental and technological applications.

## Figures and Tables

**Figure 1 sensors-21-06506-f001:**
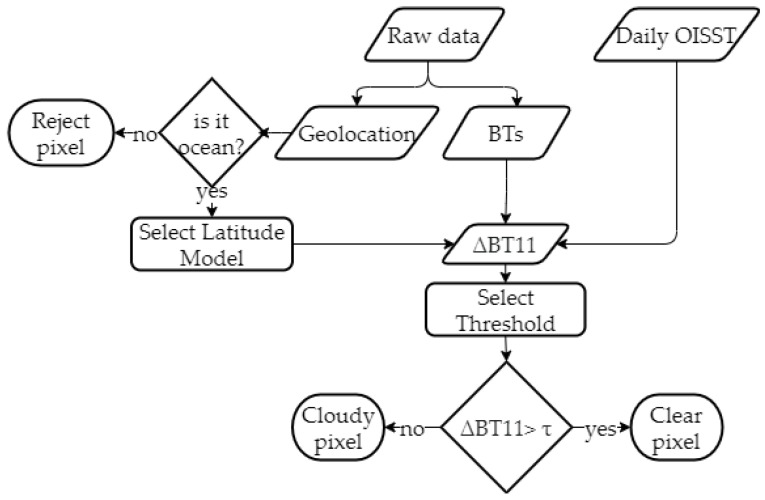
This flowchart shows the process that the SWCM follows when determining whether a pixel is clear or cloudy.

**Figure 2 sensors-21-06506-f002:**
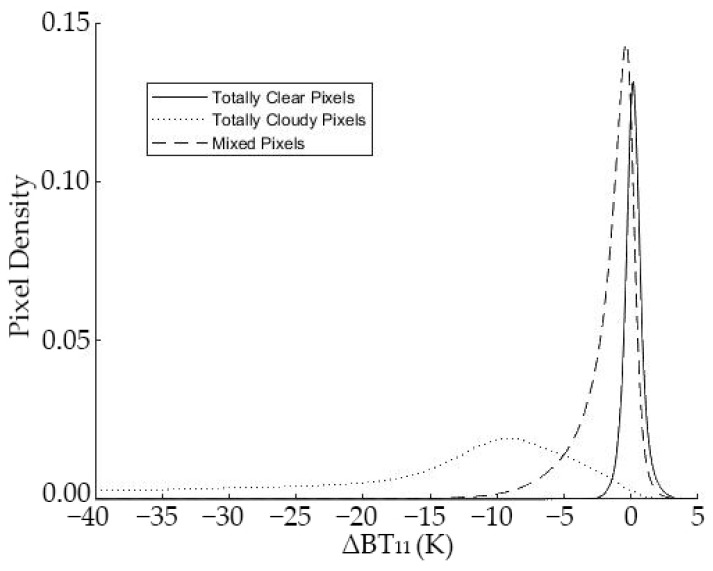
Probability density functions for clear, covered, and partially covered conditions. Midlatitude case.

**Figure 3 sensors-21-06506-f003:**
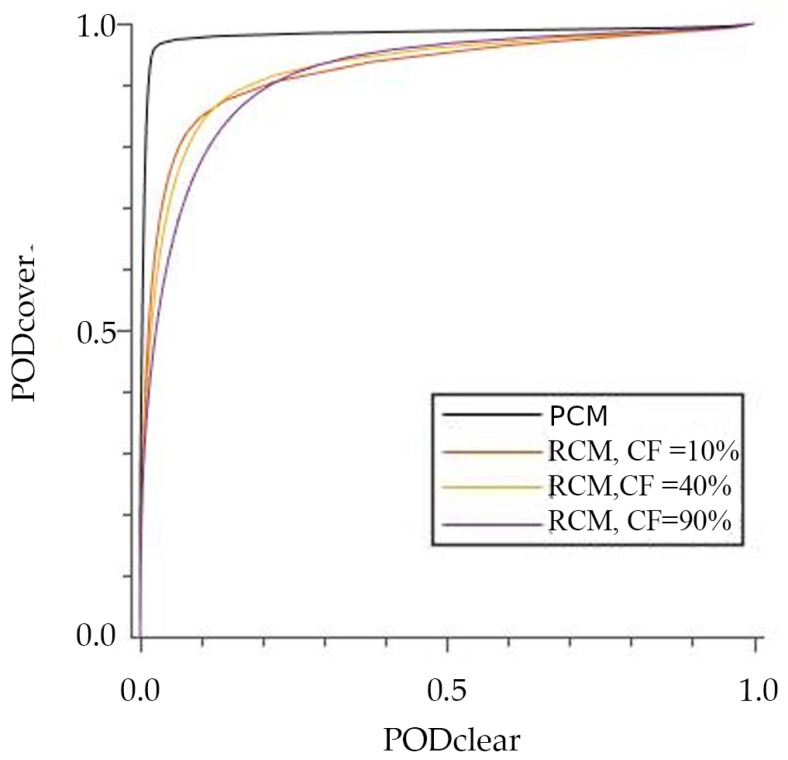
ROC curves for different reference CF thresholds.

**Figure 4 sensors-21-06506-f004:**
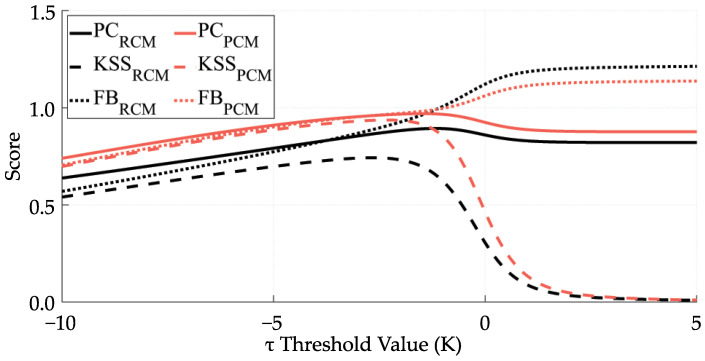
Different skills scores in function of the threshold value. midlatitude case.

**Figure 5 sensors-21-06506-f005:**
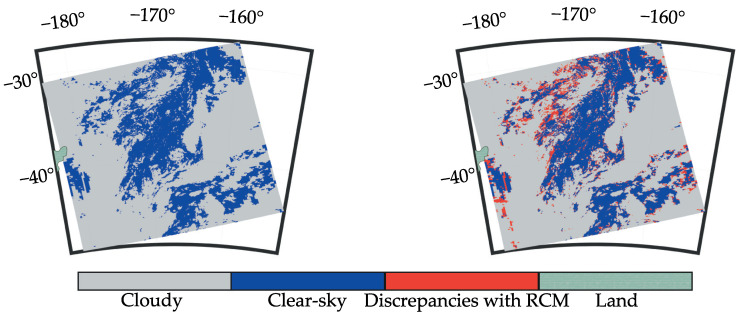
Selected case showing the performance of the mask comparing to MODIS CM. Left image shows the RCM. Right image shows the discrepancies between the SWCM and the RCM.

**Figure 6 sensors-21-06506-f006:**
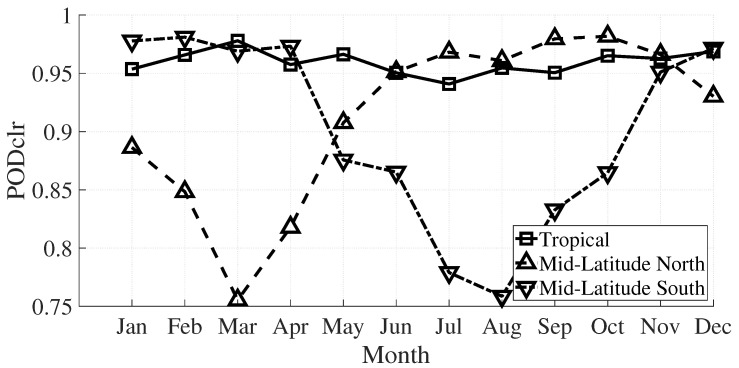
POD_clr_ in function of the month for tropical and midlatitude.

**Figure 7 sensors-21-06506-f007:**
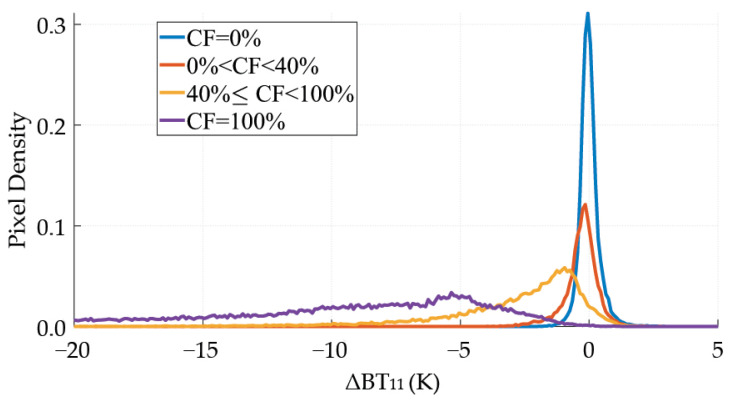
$\Delta BT_{11}$ probability density functions of different cloud fraction cases.

**Table 2 sensors-21-06506-t002:** Data used in this work.

Concept	Source	Resolution
11 and 12 μm BTs	MODIS Collection 6	5×5 km${}^{2}$
SST	NOAA Daily OISST	0.25${}^{\circ }$ × 0.25${}^{\circ }$
Cloud Fraction	MODIS Collection 6	5×5 km${}^{2}$
Zenith	MODIS Collection 6	5×5 km${}^{2}$
Geolocation	MODIS Collection 6	5×5 km${}^{2}$

**Table 3 sensors-21-06506-t003:** Calculated coefficients.

	*A*	$B_{1}$	$B_{2}\left(K^{-1}\right)$	*C*	*D(K)*
midlatitude	1.04	34.60	−0.13	1.41	−12.41
Tropical	0.95	14.28	−0.06	1.32	15.91

**Table 4 sensors-21-06506-t004:** Optimal thresholds τ values.

	Mid Latitude		Tropical	
	Day	Night	Day	Night
$\tau_{opt}$ RCM (K)	−1.7	−1.9	−1.4	−1.9
$\tau_{opt}$ PCM (K)	−1.7	−2.0	−1.8	−2.6

**Table 5 sensors-21-06506-t005:** Skills scores of the SWCM compared against the PC and RCM during day and night-time, including tropical and midlatitude.

**Tropical**						
		**RCM**			**PCM**	
	**Total**	**Day**	**Night**	**Total**	**Day**	**Night**
*a*	57,266,328	26,683,057	30,583,271	40,867,567	19,288,469	21,579,098
*b*	1,222,183	705,236	516,947	59,935	47,947	11,988
*c*	7,957,351	3,954,977	4,002,374	922,384	556,335	366,049
*d*	29,052,983	15,617,908	13,435,075	13,756,951	8,228,695	5,528,256
*n*	95,498,845	46,961,178	48,537,667	55,606,837	28,121,446	27,485,391
PC	0.90	0.90	0.91	0.98	0.98	0.99
KSS	0.84	0.83	0.85	0.97	0.97	0.98
$POD_{Cld}$	0.88	0.87	0.88	0.98	0.97	0.98
$POD_{Clr}$	0.96	0.96	0.96	1.00	0.99	1.00
$FB_{Cld}$	0.90	0.89	0.90	0.98	0.97	0.98
$FB_{Clr}$	1.22	1.20	1.25	1.06	1.06	1.06
**Tropical**						
		**RCM**			**PCM**	
	**Total**	**Day**	**Night**	**Total**	**Day**	**Night**
*a*	117,985,325	57,529,262	60,456,063	98,187,245	48,678,004	49,509,241
*b*	2,284,878	755,258	1,529,620	1,006,353	322,617	683,736
*c*	10,823,371	5,367,212	5,456,159	2,877,836	1,489,219	1,388,617
*d*	23,828,633	13,856,568	9,972,065	13,014,840	8,197,908	4,816,932
*n*	154,922,207	77,508,300	77,413,907	115,086,274	58,687,748	56,398,526
PC	0.92	0.92	0.91	0.97	0.97	0.96
KSS	0.83	0.86	0.78	0.90	0.93	0.85
$POD_{Cld}$	0.92	0.91	0.92	0.97	0.97	0.97
$POD_{Clr}$	0.91	0.95	0.87	0.93	0.96	0.88
$FB_{Cld}$	0.93	0.93	0.94	0.98	0.98	0.99
$FB_{Clr}$	1.33	1.32	1.34	1.13	1.14	1.13

**Table 6 sensors-21-06506-t006:** Seasonal variation of the obtained skills scores when comparing the SWCM against the RCM.

**Tropical**													
	**Total**	**January**	**February**	**March**	**April**	**May**	**June**	**July**	**August**	**September**	**October**	**November**	**December**
PC	0.90	0.90	0.90	0.89	0.90	0.90	0.91	0.91	0.91	0.91	0.90	0.91	0.90
KSS	0.84	0.83	0.84	0.82	0.82	0.83	0.84	0.84	0.85	0.84	0.84	0.85	0.84
POD ${}_{Cld}$	0.88	0.87	0.87	0.85	0.87	0.86	0.89	0.90	0.90	0.89	0.87	0.88	0.88
POD ${}_{Clr}$	0.96	0.95	0.97	0.98	0.96	0.97	0.95	0.94	0.95	0.95	0.97	0.96	0.97
FB ${}_{Cld}$	0.90	0.89	0.89	0.86	0.89	0.88	0.91	0.93	0.91	0.91	0.89	0.90	0.89
FB ${}_{Clr}$	1.22	1.24	1.25	1.26	1.22	1.21	1.18	1.18	1.22	1.19	1.21	1.26	1.25
**Midlatitude North**													
	**Total**	**January**	**February**	**March**	**April**	**May**	**June**	**July**	**August**	**September**	**October**	**November**	**December**
PC	0.91	0.93	0.92	0.90	0.90	0.91	0.91	0.88	0.92	0.90	0.92	0.93	0.93
KSS	0.83	0.82	0.78	0.69	0.74	0.82	0.85	0.82	0.86	0.84	0.88	0.88	0.87
POD ${}_{Cld}$	0.91	0.94	0.93	0.94	0.93	0.91	0.90	0.85	0.90	0.87	0.90	0.92	0.94
POD ${}_{Clr}$	0.92	0.89	0.85	0.76	0.82	0.91	0.95	0.97	0.96	0.98	0.98	0.97	0.93
FB ${}_{Cld}$	0.93	0.96	0.96	0.99	0.97	0.93	0.91	0.86	0.92	0.87	0.91	0.93	0.94
FB ${}_{Clr}$	1.29	1.29	1.20	1.07	1.10	1.32	1.38	1.45	1.24	1.29	1.30	1.31	1.41
**Midlatitude South**													
	**Total**	**January**	**February**	**March**	**April**	**May**	**June**	**July**	**August**	**September**	**October**	**November**	**December**
PC	0.92	0.92	0.91	0.91	0.93	0.91	0.91	0.91	0.91	0.92	0.92	0.92	0.92
KSS	0.82	0.89	0.88	0.87	0.89	0.79	0.78	0.71	0.69	0.76	0.80	0.86	0.88
POD ${}_{Cld}$	0.92	0.91	0.90	0.90	0.92	0.92	0.92	0.94	0.93	0.93	0.93	0.91	0.91
POD ${}_{Clr}$	0.91	0.98	0.98	0.97	0.97	0.88	0.87	0.78	0.76	0.83	0.86	0.95	0.97
FB ${}_{Cld}$	0.94	0.91	0.90	0.91	0.92	0.94	0.95	0.97	0.97	0.96	0.96	0.92	0.92
FB ${}_{Clr}$	1.35	1.44	1.47	1.50	1.38	1.40	1.28	1.18	1.17	1.20	1.25	1.39	1.46

**Table 7 sensors-21-06506-t007:** Other test and cloud mask results [11,39].

Different CM Products	PC	KSS	POD${}_{\mathrm{Cld}}$	POD${}_{\mathrm{Clr}}$	FAR_cld_	FAR_clr_
SWCM (Using MYD06 data)	0.91	0.84	0.90	0.94	0.02	0.27
AGRI CM Product	0.91	0.79	0.93	0.86	0.04	0.22
AHI CM Product	0.93	0.83	0.94	0.89	0.03	0.22
INSAT-3D GMM CM	0.76	0.52	0.76	0.76	0.26	0.22
$BT_{11}$ Gross Test	0.79	0.58	0.79	0.79	0.05	0.56

## Data Availability

The Aqua/MODIS Cloud Product dataset was acquired from the Level-1 and Atmosphere Archive & Distribution System (LAADS) Distributed Active Archive Center (DAAC), located in the Goddard Space Flight Center in Greenbelt, Maryland (https://ladsweb.nascom.nasa.gov).

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
