# Peer review of "Bi-Spectral Infrared Algorithm for Cloud Coverage over Oceans by the JEM-EUSO Mission Program"

_sensors, 2021, doi:10.3390/s21196506_

Round 1
Reviewer 1 Report
In this work, the authors present a method to determine the cloud map from the images provided by a bi-spectral infrared camera within the framework of JEM-EUSO (The Joint Experiment Missions - Extrem Universe Space Observatory). A test to generate a binary cloudiness mask (CM) over the ocean, employing bi-spectral IR data is presented. The CM is based on a split-window algorithm which uses an estimation of the brightness temperature calculated from a statistical study of an IR images database along with an ancillary sea surface temperature. The difference between the measured and estimation of the brightness temperature determines whether a pixel is cover or clear. Experimental results demonstrated an accuracy above 90% which is similar to that of other studies. In general, the manuscript is interesting, well organized and deserves to be published. Here some issues which I encourage the authors to consider:
- More comparisons with previous works could be a nice complement for the current manuscript. Then, the authors could demonstrate how the proposed approach outperform the current state of the art.
- There are a little grammatical/style error. In my opinion, a grammar/style revision has to be carried out before the manuscript can be considered for publication.
Author Response
Anwser in pdf.

Reviewer 2 Report
Please see the attached file.

Author Response
Answer in pdf.

Reviewer 3 Report
The authors of submission sensors-1327094 present an interesting work on cloud detection. They present a method to determine a map of clouds from the images provided by a bi-spectral infrared camera, which is very important, for example, in climate study. A literature review, as well as theory introduction, is extensive, reliable and solid. In the methods section, they described their proprietary approach called Split-Window Cloudiness Mask (SWCM) derived from the split-window algorithm. They describe how to design and validate SWCM, as well as the methodology to optimize the thresholds that characterize SWCM. The authors claim that cloudiness mask will be applied in the future to JEM-EUSO systems, so it must be based on the data provided by those systems. However, the methodology description that they provided is too vague and should be clarified. Would you mind trying to revision section two and giving only the necessary information? In the present form, it gives an impression of chaos. I suggest presenting some flowcharts and make them more digestible. Even a reader highly skilled in the topics you describe may have difficulty understanding whether the actions you are taking are needed, or at best, will have to spend a lot of time on it. The results of this work have been split into two subsections, first specifies the results of the proposed methodology, that is, the coefficients and thresholds of the SWCM. The second part presents the validation of the results. They also deliberate on how the day-night cycle and seasonal variation affect the results. The coefficients that they have calculated are valid for the tropical and midlatitude models. Similar coefficients characterize both models except for two of them associated with the Brightness Temperature Difference. Finally, they obtained eight different thresholds. The authors provided a discussion part in which they compared obtained results with other authors results. The conclusion part needs to be extended. The paper has 19 pages where the conclusion part is less than one page. The topic is of interest in that it has the potential to be used in engineering practice. Overall, the paper is well-structured but illegible and not coherent. The work contains a lot of information that can be structured better. Several points need to be clarified. I would recommend a major revision.
General comments
- The abstract should be rewritten. It is not clear what is your unique contribution;
- There are many different abbreviations, to improve readability, I suggest explaining these abbreviations in the appendix;
- The paper contains some stylistic and structural errors. It would be best to try to avoid wordiness and limit the use of passive voice to improve readability. I would recommend proofreading;
- Your paper lacks a proper summary. Consider adding such a section. You can present it as "Summary and Discussion";
- You should extend the "Conclusions part" with a focus on your unique contribution. It is an essential part of the manuscript.
- You presented a lot of information, are they all needed to evaluate the suitability of your method?
Additional questions
- Could you describe the boundary conditions of the SWCM method?
- Could you deliberate more on the resilience of the SWCM method?
Author Response
Answer in pdf.

Round 2
Reviewer 2 Report
All my concerns have been solved.
Reviewer 3 Report
Thank you for addressing all my comments and questions. I believe that the manuscript has been significantly improved.
This manuscript is a resubmission of an earlier submission. The following is a list of the peer review reports and author responses from that submission.
Round 1
Reviewer 1 Report
The paper designs and evaluates a CM test to determine the presence of clouds in the FoV of the JEM-EUSO telescope, over oceans and during nighttime. It uses a method based on thresholding and optimizing skill scores to separate images. However, I felt such a method is not novelty enough since nowadays so many learning-based methods can handle this task efficiently and robustly. At least the authors should try some learning-based methods or give the reason why they haven't consider this issue.
The format of the manuscript needs to be adjusted.
Some typos, fonts, notation problem:
line 113, 267, 370
Reviewer 2 Report
I am sorry to have to express such a harsh opinion, but I consider this manuscript to be somewhat chaotic, from the abstract to the conclusions. In addition, the style and grammar of written English add to the confusion. In general, the structure, although it has the typical sections of a research article, when you read each of them carefully, there is a continuous repetition and justification of the study from the Introduction to the discussion.
Concerning the Introduction, it cannot be considered as such. It does not discuss the state of the art at any point. No other similar studies are considered and no references are added. This section is a kind of extended summary of the whole manuscript, where the methodology and the results obtained are presented. The rest of the manuscript is not organised in a natural and orderly way, describing first the data or sources of information used to make their estimates and then the method used to construct their algorithm. It is not until line 268 that they present the steps of their methodology in summary form. This should have been stated from the beginning, to aid the reader's understanding. Even in the results section, the authors add new methods and calculations that were not discussed before.
I will not go into the numerous errors in writing, references, etc., throughout the manuscript. But let the first words of the title serve as an example: "Bi-Spectral Infrared Cloud Coverage Algorithm ... ". Would it not have been more appropriate, to avoid confusion, to put "Bi-spectral infrared algorithm for cloud coverage over oceans..."?
Honestly, I don't see that this study brings any novelty or improvement in this field of knowledge. At least, as it has been presented. The results do not seem to be of interest to the average Sensors reader. Not even, dare I say, for a reader accustomed to this methodology and data from a journal of the same publisher such as Remote Sensing.
I encourage the authors, in any case, to re-organise the manuscript, revise the English extensively and re-submit it with a more appropriate approach for this journal.
Here are some general recommendations for the authors to redirect this study. They should give much more importance to the characteristics of the sensor used and to what extent it is comparable to the MODIS sensor channels 11 and 12 they put forward. The authors would include relevant information on the filter functions of the channels and their equivalence to those of MODIS. They would also discuss the problem of using as a sea temperature reference a product such as OISST, which is derived from a model that uses satellite data but also in situ buoy measurements, and which is in no way equivalent to the skin SST derived from a remote sensor. They would also omit all the theory relating to the development of split-window algorithms, which is well known to an interested reader and can be found in any textbook in undergraduate and postgraduate courses.